# An Agent-Based Model of Radiation-Induced Lung Fibrosis

**DOI:** 10.3390/ijms232213920

**Published:** 2022-11-11

**Authors:** Nicolò Cogno, Roman Bauer, Marco Durante

**Affiliations:** 1Biophysics Department, GSI Helmholtzzentrum für Schwerionenforschung GmbH, 64291 Darmstadt, Germany; 2Institute for Condensed Matter Physics, Technische Universität Darmstadt, 64289 Darmstadt, Germany; 3Department of Computer Science, University of Surrey, Guildford GU2 7XH, UK

**Keywords:** agent-based modelling, RILF, IPF, senescence, bystander, 3D modelling, NTCP

## Abstract

Early- and late-phase radiation-induced lung injuries, namely pneumonitis and lung fibrosis (RILF), severely constrain the maximum dose and irradiated volume in thoracic radiotherapy. As the most radiosensitive targets, epithelial cells respond to radiation either by undergoing apoptosis or switching to a senescent phenotype that triggers the immune system and damages surrounding healthy cells. Unresolved inflammation stimulates mesenchymal cells’ proliferation and extracellular matrix (ECM) secretion, which irreversibly stiffens the alveolar walls and leads to respiratory failure. Although a thorough understanding is lacking, RILF and idiopathic pulmonary fibrosis share multiple pathways and would mutually benefit from further insights into disease progression. Furthermore, current normal tissue complication probability (NTCP) models rely on clinical experience to set tolerance doses for organs at risk and leave aside mechanistic interpretations of the undergoing processes. To these aims, we implemented a 3D agent-based model (ABM) of an alveolar duct that simulates cell dynamics and substance diffusion following radiation injury. Emphasis was placed on cell repopulation, senescent clearance, and intra/inter-alveolar bystander senescence while tracking ECM deposition. Our ABM successfully replicates early and late fibrotic response patterns reported in the literature along with the ECM sigmoidal dose-response curve. Moreover, surrogate measures of RILF severity via a custom indicator show qualitative agreement with published fibrosis indices. Finally, our ABM provides a fully mechanistic alveolar survival curve highlighting the need to include bystander damage in lung NTCP models.

## 1. Introduction

Radiation-induced lung fibrosis (RILF) and idiopathic pulmonary fibrosis (IPF) are chronic lung diseases characterized by progressive stiffening of the alveolar walls leading to respiratory failure and eventually death [1,2,3,4]. Despite sharing multiple pathways and a unique endpoint, IPF and RILF have different roots: while the former does not stem from a single known cause, the latter is the final stage of a causal sequence triggered by intracellular and extracellular ionizations.

Regardless of the damage source, alveolar epithelial cells of type 2 (AEC2) play a central role as triggers of both IPF and RILF [3,5,6,7,8,9,10,11,12]. Regarded as progenitor/stem cells, AEC2 are capable of self-renewal [12] and act as a reservoir for the terminally differentiated alveolar epithelial cells of type 1 (AEC1), which are responsible for the gas–blood exchange [13]. Irreparable injury to AEC2 is resolved either through apoptosis or a switch to a senescent phenotype [14] that allows AEC2 to secrete chemokines and cytokines that activate the immune response. Among those, the monocyte chemoattractant protein 1 (MCP1) draws the monocyte-derived macrophages (M0) into the alveolar lumen, where they differentiate into M1 macrophages [4,9]. The latter can then differentiate into pro-fibrotic macrophages M2, which modulate the proliferation of mesenchymal cells. By secreting factors such as transforming growth factor beta (TGFβ), platelet-derived growth factor (PDGF), and interleukin 13 (IL13), M2 stimulates fibroblasts’ proliferation and differentiation into myofibroblasts [15,16,17,18]. Finally, extracellular matrix (ECM) secreted by both fibroblasts and myofibroblasts stiffen the alveolar walls leading to lung scarring, which hinders breathing [19,20] (see our previous work for more details [21]).

RILF and radiation pneumonitis constrain the maximum dose and irradiated volume in thoracic radiotherapy (RT). Despite RT’s extensive usage and proven effectiveness in cancer therapy, radiation-induced lung injuries (RILI) among lung cancer patients have been reported with a 7 to 30% incidence [22,23,24]. Normal tissue complication probability (NTCP) models are used to predict the risk of undesired endpoints given a patient’s treatment plan [25] and can be classified into phenomenological, which rely only on clinical data, and biological, which take into account previous knowledge in the tissues’ morphology [26]. Among the latter, the concept of functional subunits (FSUs) is widely adopted [26,27,28]: tissues are thought of as clusters of independent entities (the FSUs) capable of performing a specific organ function that can be fully regenerated by a single stem cell. The conceptual arrangement has been adopted both for serial organs, such as the spinal cord, and parallel organs, such as the lungs, for which both the alveolus and the acinus have been proposed as FSUs candidates. Among the biological NTCP models, the critical volume model [27] states that normal functioning of organs and tissues is not impaired by radiation unless the fraction of surviving FSUs falls below a certain threshold. Consequently, the critical functioning volume that must be spared to avoid complications is determined by the size of the threshold fraction. Given that an FSU can be fully restored by a single surviving clonogenic cell, the probability that an FSU is depleted corresponds to the probability that all the stem cells that belong to it (e.g., the AEC2 in the alveoli) are killed.

Although the current biological NTCP models adopt a priori non-clinical knowledge to estimate patients’ risk, they lack a mechanistic representation of the undergoing processes. The biological background is limited to morphological information on the tissue (encoded in the FSUs size, shape, and number) supplemented uniquely by data on cells’ radiosensitivity. The latter, combined with the delivered dose, provides an estimate of the surviving cells. Mechanisms such as indirect damage spread by irradiated cells, cell repopulation, and repair kinetics are neglected by the current models, and the picture of the radiation-induced injury progression is, thus, incomplete. In this regard, a contribution might be given by agent-based models (ABMs), in which autonomous entities (such as cells) are assigned behaviours and states and left free to interact with each other and with the surrounding environment [29]. According to this paradigm, agents take decisions that depend on probabilistic rates and often lead to unpredictable emergent collective phenomena. Unlike equation-based models (EBMs), which describe the dynamics of population averages, the decentralized approach offered by ABMs allows for individual tracking and local information exchange, providing more realistic models [30].

We implemented a 3D ABM of a human alveolar duct with cell resolution (see Figure 1) that simulates the onset of RILF and its progression over the course of 3 years.

The model builds on our previous work [21] with an emphasis on cell senescence and cell repopulation. A finer spatial resolution allows for radiation-induced damage to AEC2 to spread to bystanders [12,31], while macrophages can phagocyte senescent cells [32,33]. The grid arrangement of the simulation space constrains local impact on the surrounding environment by the cells and enables heterogeneous damage distributions. Patterns of ECM accumulation from our model match experimental findings of early and late computed tomography (CT) changes in lung cancer patients treated with RT [34], and we show total RILF resolution for low doses as reported by previous studies on mice cells [5]. Moreover, the dose-response curve for the ECM accumulation from our model matches the sigmoidal responses reported in the literature [35,36]. We used the number of healthy AEC2 per alveolus to track FSU survival and observed a divergence from the theoretical values predicted by the linear-quadratic (LQ) model of cell killing adopted by current biological NTCP models [26]. Based on this, we stress the need to account for microenvironmental factors when estimating the risk of RILI. Finally, we combine the ECM dose-response curve and the alveoli survival into a custom indicator for the surrogate measure of RILF severity and show that it is in agreement with experimental data from mice [37].

## 2. Results

### 2.1. Early and Late Fibrotic Response

We investigated the effect of the following model parameters on the outputs of the model: (i) the fraction of phagocytic macrophages, (ii) the AEC2 apoptotic-to-senescent ratio, and (iii) the bystander senescence threshold. More specifically, we tracked the number of cells and the concentration of the extracellular substances for 1000 days to ensure the stabilisation of the system (either towards RILF resolution or sharpening). Figure 2 and Figure 3 show the total cell number and average substances concentration for different doses (which correspond to an average AEC2 survival in the range 95% to 1%) with phagocytic fraction = 100% (i.e., all the macrophages were able to phagocyte 1 senescent cell), apoptotic-to-senescent ratio = 0 (i.e., all the damaged cells switched to the senescent phenotype and none underwent apoptosis) and bystander threshold = 2 (i.e., at least 2 senescent cells in the neighbourhood of a healthy AEC2 were needed to induce damage). Following a sharp rise in the number of damaged (and then senescent) AEC2, the immune system is triggered, and the senescent AEC2 are fully cleared within one year at all the doses. Within the alveoli irradiated at or below 4-5 Gy, the surviving healthy AEC2 are able to regenerate the whole epithelium (both AEC1 and AEC2), while for higher doses, there is an increase in the number of alveoli with no surviving stem cells. Consequently, both the M1 and M2 populations are reduced to their homeostatic levels. Due to the absence of healthy AEC2, the mesenchymal cells proliferate and secrete ECM that settles, with concentration proportional to the damage, after an initial rise.

In what follows, the term ECM concentration will refer to the average extracellular matrix concentration across the whole simulation space in g/cm^3^, used as a surrogate of the more widely adopted Hounsfield Units. Figure 4 shows the early fibrotic response, that is, the absolute increase in the ECM concentration (with respect to the homeostatic values) at multiple doses 3 months after the irradiation. We separated the early component from the late one as was conducted in previous analyses [34] and fitted the experimental data using a logistic function as shown in [35,36,42] (we replaced the Hounsfield units (HU) with the ECM concentration):(1)$$\Delta \mathrm{ECM}=\frac{\Delta ECM_{max}}{1+e^{4*\gamma *\left(1-\frac{D}{D_{50}}\right)}}$$
where $\Delta ECM_{max}$ is the maximum ECM concentration, γ is the steepness of the sigmoid and $D_{50}$ is the dose at 50% of the maximum ECM. An increase in the ECM concentration is observed even at very low doses (<2 Gy), while saturation occurs above 12 Gy.

When lowering the fraction of phagocytic macrophages (while keeping the apoptotic-to-senescent ratio and the bystander threshold constant) to 60% and 80%, we observed the saturation of the ΔECM at 4 Gy and 6 Gy, respectively.

We measured the average increase in the ECM concentration at 1000 days from the irradiation for multiple doses and fitted the experimental data with the probit f unction provided in [37]:(2)$$\Delta \mathrm{ECM}=\frac{1}{2}*A*\left\{1-erf\left(\sqrt{\pi }*\gamma *\left(1-\frac{D}{ED_{50}}\right)\right)\right\}$$
where *A* is the maximum ΔECM concentration, *erf* is the error function, *γ* is the maximum steepness, and *ED*_50_ is the dose at 50% of the maximum ΔECM concentration. As reported in multiple studies [34,35,43], we assumed negligible ECM changes at late time points for doses < 5 Gy and excluded those data from the experimental dataset. Given that the maximum dose (17.6 Gy) damaged 99% of the healthy AEC2, higher doses would not result in further increases of the ΔECM, and, therefore, we assume that the measured ΔECM_max_ is close to its saturation value. Overall, we observed a good agreement between the experimental data, shown in Figure 5, and the fitting curve. Although both the early and late fibrotic responses could be fitted by a sigmoid, we observed the halving of the maximum ECM concentration in the late phase for the doses that induced RILF. Finally, we measured *A* = 0.0030 and *A* = 0.0040 for the 60% and 80% phagocytic fraction, respectively.

### 2.2. Alveoli Survival

We investigated the survival of the alveoli 1000 days after the irradiation by counting the number of surviving healthy AEC2 per alveolus. We made the assumption that only AEC2 can be damaged by irradiation, and damaged cells can either undergo apoptosis or become senescent. In either case, cells lose their clonogenic potential. Further, for the sake of simplicity, we did not take into account the severity of the damage or repair mechanisms; thus, a cell can either be damaged or healthy. As shown in Figure 3, all the senescent AEC2 were removed by the immune system within one year and, therefore, each alveolus that retained at least one healthy AEC2 restored the whole epithelium and did not differ from an irradiated one when the measurement was performed. As reported by previous studies, each alveolus in our model represented an FSU that could be fully regenerated by the AEC2 acting as stem cells. Building on the concepts outlined in the critical volume NTCP model [26,27], where the survival probability of an FSU is given by the probability that at least one of its stem cells survives, as shown by Equation (3), we fitted the experimental data on the surviving fraction of the alveoli using the aforementioned LQ model, which was adopted to estimate the survival of a single cell when irradiated at the dose *D*.
(3)$$P_{surv,FSU}=1-P_{kill,FSU}=1-\overset{N_{AEC2}}{\underset{i=1}{\prod }}P_{kill,cell}=1-{\left(P_{kill,cell}\right)}^{N_{AEC2}}\;=1-{\left(1-e^{-\alpha D-\beta D^{2}}\right)}^{N_{AEC2}}$$

As can be seen in Figure 6, the LQ curve with *α*/*β* = 2 is not in agreement with the experimental data and would predict lower survival for the doses at the end of the simulated range.

### 2.3. RILF Severity Index

As seen in a previous study on RILF in mice [37] where a fibrosis index (FI) was introduced, we implemented our own surrogate measure of RILF severity, the RILF severity index (*RSI*). To make comparisons between the RILF and the FI reasonable, we defined the *RSI* as follows:(4)$$RSI=\sqrt{\Delta \bar{ECM_{conc}↑}*\Delta V_{surv,FSU}↓}$$
where $\Delta \bar{ECM_{conc}↑}$ is the average increase in the ECM concentration across the whole simulation space (in g/cm^3^) and $\Delta V_{surv,FSU}↓$ is the decrease in functioning distal lung volume given the total volume of the surviving FSUs (in cm^3^). The experimental data are shown in Figure 7 and were fitted using the equation provided in [37] and presented in Section 2.1 (Equation (2)). Again, we assumed negligible ECM changes at late time points for doses < 5 Gy and excluded those data from the experimental dataset.

Figure 7 shows good agreement between the experimental data and the model curve; thus, the *RSI* matches the *FI* quantitatively up to a proportionality constant. We expect parameter *A* to saturate for *D* > 17.6 Gy, which would lead to 0% FSU survival.

### 2.4. Effects of AEC2 Apoptotic to Senescent Ratio

Previous studies [5,11] have attributed the onset of RILF and IPF to the emergence of the senescent phenotype among the AEC2 rather than their apoptosis. Consequently, we investigated the impact of different values of the AEC2 apoptotic-to-senescent ratio on the survival of the alveoli. Given a certain amount of damaged cells, we lowered the fraction of senescent AEC2 and increased the apoptotic one, and the results are shown in Figure 8.

As expected, due to the ability of the healthy AEC2 to repopulate the depleted alveoli and the lack of damage-spreading cells, we observed larger fractions of surviving alveoli for lower AEC2 senescent-to-damaged fractions. Notably, full resolution of the RILF was achieved for ratios ≤ 50% and dose ≤ 10 Gy.

## 3. Discussion

RILF imposes strong constraints on the doses that can be delivered to thoracic tumours, limiting the potential benefits of radiation therapy. Understanding its pathophysiology and estimating the probability that normal tissue complications arise, given a combination of dose and irradiated volume, is of paramount importance. Moreover, given the similarity between RILF and IPF, the latter would benefit from further knowledge of the progression of the former. Current NTCP models are built on a phenomenological basis and rely on clinical experience to determine the tolerance doses for the organs at risk. The lack of a mechanistic description of the undergoing phenomena might lead to oversimplified models of the radiation-induced injuries featuring restricted sets of input parameters and being unable to simulate the effects of post-irradiation treatments.

In this study, we implemented an ABM of a human alveolar segment and simulated the onset of RILF after the irradiation with single-fraction X-rays. We downscaled the model presented in our previous work [21] to the cellular level and added key cellular behaviours that have been documented both in RILF and IPF, such as cell repopulation, senescent cells clearance, and bystander senescence [31,44,45]. Our study emphasizes the role of cellular and molecular mechanisms in NTCP modelling and shows the time evolution of the RILF status. After testing several combinations of the bystander threshold parameter (i.e., the minimum number of senescent neighbours needed to trigger the phenotypic switch for a healthy AEC2) and phagocytic fraction (i.e., the fraction of macrophages able to remove senescent cells), we set them to 2 and 100%, respectively. When using these values, we observed full clearance of the senescent cells and epithelial recovery at doses < 5 Gy, as shown by Citrin et al. [5].

Our model, consisting of 18 alveoli organized in a cylindrical shape and 8 different cell types, was calibrated in homeostatic conditions using the results from the IPF model by Hao et al. [46]. Radiation-induced damage was simulated as a shift, for the AEC2, from the healthy to the senescent phenotype, with higher doses corresponding to larger damaged fractions. Experimental data from a study on the survival of mice AEC2 [47] were fitted using the LQ model curve that provided a dose–survival relation. By means of that, we simulated the irradiation of the alveolar duct within the dose range [0.6–17.6] Gy (which corresponds to 95% to 1% cell survival), and we tracked the total number of cells for each cell type and the substances concentration for 1000 days following the irradiation. In particular, the increase in the average ECM concentration and the decrease in the number of epithelial cells (both AEC1 and AEC2), hallmarks of tissue stiffening and inability to recover, respectively, were adopted as measures of RILF severity. A significant increase in the ECM concentration was observed at 3 months following the irradiation due to the spread of the senescent AEC2 and the consequent rise in the number of mesenchymal cells. The subsequent clearance of the senescent cells from the macrophages reduced the inflammation, and the homeostatic conditions were restored for the alveoli irradiated at doses < 5 Gy circa. Although all the senescent AEC2 were eventually removed at each simulated dose, some alveoli (where the number increased with the dose, as will be detailed in the next section) were fully depleted from the epithelial cells and underwent a permanent increase in the ECM. Our results, showing two distinguishable fibrotic responses, an early and a late one, are consistent with patterns detected in human lungs irradiated with photons [34,43,48] where density changes with respect to the baseline were measured at different time points. Moreover, the time-to-peak for the early response matches quantitative experimental data reported in the literature [34,35]. Finally, as shown in Figure 4 and Figure 5, a good agreement was observed between the increase in ECM concentration as a function of the delivered dose and the models provided by [35] and [37] for the early and late response.

Building on the concept of the FSUs, widely used in modern biological NCTP models such as the critical volume model [26,27], we measured the alveoli survival for multiple doses at 1000 days from the irradiation. Specifically, we identified the alveoli as FSUs of the lung in our model and classified as killed the alveoli fully depleted from the healthy AEC2. We fitted our data using the survival curve used in the critical volume model, which assumes that the probability that an FSU is killed is given by the joint probability of having all the stem cells killed, where the survival of a stem cell follows the LQ model. Figure 6 shows discrepancies between our data and the given model, with lower survival predicted by the model for the doses at the higher end of the simulated range. We suppose that the cause of this difference is the AEC2 repopulation not being compensated by the bystander senescence mechanism. Consequently, we argue that the temporal evolution of the RILF should play a role when modelling NTCP, as the FSUs survival might not be fully described by the LQ model alone.

We introduced a custom indicator, the *RSI*, as a surrogate measure of the RILF severity. Based on the idea of the fibrosis index implemented by Zhou et al. [37], the *RSI* is a positive number that measures the fibrosis damage by combining the depletion of the alveoli (and therefore AEC2 and AEC1) with the increase in the average concentration of the ECM. For each dose > 5 Gy, we computed the reduction in the functioning lung tissue as the loss of the volume due to the depleted alveoli and measured the average rise in the ECM concentration with respect to the homeostatic values at 1000 days from the irradiation. The results, which we fitted using the equation provided in [37], show quantitative agreement with the model parameters *γ* and *ED*_50_ (while the fibrosis index differs by a multiplicative constant). Taken together, these outcomes support the *RSI* as an alternative to the FI for computational simulations where a direct measure of the decrease in the functioning volume might not be available.

To investigate the impact of the senescent AEC2 on the RILF severity, we measured the alveoli survival with multiple senescent-to-damaged ratios when irradiated at different doses. Despite the increase in the number of apoptotic AEC2 and, as expected, lowering the fraction of senescent AEC2 from 100% to 20% increased the alveoli survival from 33% to 72% at 13.7 Gy. Notably, this feature could be exploited to simulate the effects of senolytic drugs on lung tissue, which have been shown to be able to reverse persistent RILF and IPF in mice [49,50,51]. The accumulation of senescent cells with age in the human lungs [52,53], along with the finding that ageing is among the major risk factors for RILI [24,54,55], further support the relationship between the senescent AEC2 and RILF severity. Our work could therefore set the basis for modelling and testing the modulation of NTCP when RT is combined with drugs ([56] is an example of an ABM where the impact of chemotherapeutic drugs is simulated). Such models could then be exploited to optimize the treatments as a step towards personalized medicine.

This study showed that the model could replicate results published in the literature and highlighted discrepancies between the alveoli survival predicted by the critical volume model and our simulations. Contextually, we draw attention to the importance of both cell–cell and cell–environment interactions which introduce a time dependence in the survival curve of the alveolar stem cells.

## 4. Materials and Methods

We developed a hybrid ABM of an alveolar segment in a three-dimensional space that simulates cellular dynamics and molecular signalling following radiation-induced damage. Our model focuses on cell-scale behaviours and provides insights into RILF and normal-tissue complication modelling. In the following sections, we provide a detailed overview of the model implementation and present its main features. For a complete list of the model’s parameters, see the Appendix A.

### 4.1. Software Platform and Modelling Environment

Our model was developed using BioDynaMo (BDM, version v1.03.15-9589c6e2, Copyright (C) 2022 CERN & University of Surrey for the benefit of the BioDynaMo collaboration. All Rights Reserved), an open-source framework for building biological ABMs [57] in C++ (the source code of our model is provided in the Appendix A). Prototype simulations were performed on a MacBook Pro 2018 running macOS Monterey on a 2.3 GHz Quad-Core Intel Core i5 processor with 8 GB RAM, while a compute node of the Lichtenberg HPC system running CentOS 8.2 on 2 × 2.3 GHz Intel Cascade-Lake AP 48-cores processor (96 total cores) with 384 GB RAM was used for simulations requiring more than one day and testing multiple parameters.

The PDEs describing the diffusion of the extracellular substances were solved using the forward Euler in time and central in space method (FTCS), while boundary conditions were set to “closed” to mimic an insulated environment. To account for the depletion of interacting substances (such as the ECM, matrix metalloproteinases (MMP), and tissue inhibitors of metalloproteinases (TIMP)), the BDM FTCS algorithm was refined, and binding coefficients were included. The simulation space was defined as a (2000 × 2000 × 2000) μm^3^ cube to fit an alveolar duct, and the 3D diffusion grid was split into cubes of side 500 μm to match the typical size of a CT voxel. The simulation time step was set to 1 s to ensure that the Courant–Friedrichs–Lewy stability condition was satisfied for the diffusion coefficient of each substance involved in the simulations. To reduce the total simulation time, the frequency of the default BDM standalone operations that regulate mechanical interactions among cells and cell behaviours was set to 10 simulation time steps. All the outputs derived from the simulations (as described in the following section) were printed on ROOT files and later on analysed using custom Python scripts (see, for example, [58]).

### 4.2. Geometric Frame and Operations

The lungs’ airway tree culminates in the acinar airways, which are responsible for the gas–blood exchange [59]. Within them, independent structures named acini bifurcate into alveolar ducts (or segments), where stacked alveoli are arranged in cylindrical shapes. To investigate the impact of cell-scale behaviours such as senescent cell clearance, bystander damage, and repopulation, we downscaled our previous model [21] and simulated a human alveolar duct where spherical agents represent alveolar cells. Alveolar ducts can be seen as hollow cylinders whose walls are lined with alveoli [59]. These latter have been previously described as three-quarter spheres with an opening that faces the lumen of the ducts. As detailed in our previous work, given the average radius of an alveolus (approximating a sphere) and the average length and diameter of an alveolar duct, we implemented an 18-alveoli model of alveolar duct, where 3 rings of 6 tangent alveoli are stacked atop one another (sizes are given in Table 1). At the beginning of each simulation, newly created agents are assigned a type and randomly distributed on the 18 spherical surfaces. Our model features 3 major cell categories which inherit methods and properties (such as the identifier of the alveolus they belong to) from a common alv_cell type. Simulated cells are categorized as follows:

Epithelial cells are positioned one alveolar radius from the alveolus centre, while mesenchymal cells and macrophages are located slightly further from (in the interstitial space) and slightly closer to (in the alveolar lumen) the alveolar centre, respectively.

As mentioned in the previous section, the simulation space is partitioned into cubic voxels that make up the diffusion grid. We provided all the cells in our model with a function that allows them to measure the concentration of all the substances within the voxel where they reside. We used this function to both inform the cells about changes in the substances’ concentration (to which they might react) and exploit the cells as probes to constantly track the average concentration of all the substances across the whole simulation space (for a complete list of the substances involved in the simulation as well as diffusion constants, decay and binding coefficients, see our previous work). To do this, and to record the total number of cells of each type, we used the update time series operation provided by BDM with a frequency of 1 h. Moreover, we developed custom standalone operations with a 10 s frequency to:determine the number of surviving alveoli;simulate the secretion of IL13 from lymphocytes (which, for simplicity, are not included in the model) at the centre of each alveolus [61];simulate the inflow of fibroblasts in the interstitial space with a rate that depends on the number of AEC2 per alveolus. According to the literature [62,63], prostaglandin E2 secreted by AEC2 inhibits fibroblasts’ chemotaxis and proliferation. We modelled this phenomenon using a reverse Hill function with coefficient 1 (to avoid abrupt changes in the fibroblasts’ flow) multiplied by a constant rate. We assumed the Hill constant to be equal to the initial number of AEC2 per alveolus to reduce the number of unknown parameters and tuned the constant influx rate to keep the number of fibroblasts in homeostatic conditions constant;simulate the flow of monocytes into the alveolar space. The rate of monocytes entering each alveolus is constant in homeostasis and is incremented in the presence of senescent AEC2, which secrete MCP1 [64] (and is measured locally). See our previous work for more details [21].

### 4.3. Cell Behaviours

Cells’ actions can be encoded and grouped into agents’ behaviours which in turn are triggered by external signals or executed at a specific rate. Depending on their type, cells in our model are assigned custom behaviours when they are added to the simulation and sequentially perform the prescribed activities until they are removed. In what follows, we provide an overview of all the behaviours implemented in our model and emphasize the major changes with respect to our old work [21]. As mentioned before, biological behaviours are executed with a frequency of 10; that is, 1 biological time step is equal to 10 diffusion time steps.

#### 4.3.1. Secretion

Cells secrete extracellular substances by changing their concentration within the voxel where they are located. The amount released by a secreting cell at each iteration can be constant or depend on external factors as detailed in what follows (see [21,46] for a deeper overview of the mechanisms):M2: PDGF (constant rate), MMP (constant rate), TIMP (constant rate), IL13 (constant rate), TGFβ_active_ (rate is positively affected by IL13 concentration);M1: TNFα (constant rate);F: TGFβ_inactive_ (constant rate), ECM (rate is affected with inverse proportionality by ECM concentration; no secretion if the concentration exceeds a saturation threshold);MF: ECM (rate is positively affected by TGFβ_active_ concentration);Senescent AEC2: TNFα (constant rate), MCP1 (constant rate), FGF2 (rate is positively affected by TGFβ_active_ concentration).

#### 4.3.2. Damage Spreading and Clearance

The central role played by senescent cells in RILI and IPF has been strongly emphasized in multiple studies [3,5,6,7,8,9,10,11,12]. More specifically, senescent AEC2 are involved in both the initiation and spread of the inflammatory process that, if unresolved, triggers the fibrotic process [10]. Unlike our previous work [21], in which the number of activated (i.e., senescent) cells damaged at the beginning of the simulation remained constant, here, the fraction of senescent cells is modelled as dynamic, both in size and position. As with multiple behaviours described in the following sections, downscaling our previous model allowed us to investigate this process.

Due to their high radiosensitivity and key role as stem cells (AEC1 are not able to differentiate [65]), we assumed that only AEC2 could be damaged by the irradiation. Irradiated cells in our model are either removed from the simulation (to simulate their apoptosis which has been proved to increase only in the first few weeks after the irradiation [5]) or switch temporarily to the damaged state [14], the duration of which depends on a random variable. Damaged cells, although slowly migrating (see the section Migration), cannot spread the damage but become able to do so as they turn senescent. In fact, senescent AEC2 acquire a senescent-associated secretory phenotype (SASP) through which they are able to trigger the immune system and damage neighbouring (bystander) healthy cells, known as the senescence-induced senescence [31]. Our implementation of the bystander senescence behaviour is based on the model of radiation-induced intercellular signalling by McMahon et al. [66], in which irradiated cells can induce damage to healthy ones by secreting extracellular mediators. Although multiple studies have reported the involvement of TGFβ_active_ in the process [67,68,69,70], we did not make any assumptions on the signalling molecules and instead modelled the intensity of the signal more generally using the intercellular distance. As in [66], our implementation relies on a threshold mechanism: when the number of senescent cells in the neighbourhood of a healthy one exceeds a threshold (which was tuned together with the senescent clearance mechanism described below to ensure complete recovery for low fractions of damaged cells), the “time above threshold” *τ* of the healthy cell is increased by one unit and decreased otherwise (if greater than 0). Since the neighbourhood of the healthy cell is defined as the group of cells within a two AEC2 diameter distance (to simulate short-range interaction), but not limited to the same alveolus, this mechanism allows for both intra- and inter-alveolus damage spreading. Finally, for a healthy cell with *τ* > 0, the probability of transitioning to the damaged type is given by:(5)$$p=1-e^{-k\tau }$$
where *k* is a cell-line-specific constant. In our model, we used the value provided in [66] for the non-small-cell lung cancer H460 cell line.

Alongside the damage spreading mechanism, in this model, we introduced a behaviour to allow the immune system to perform senescent cell clearance [32,33]. In particular, we provided the macrophages (both the M1 and M2 type, as their role in the process has not been fully elucidated yet [71]) with the ability to phagocyte senescent cells. At each time step, macrophages check whether they are in physical contact (a distance which we assumed to be 10% larger than the cell-to-cell distance) with any senescent AEC2 in the same alveolus and, if so, they can phagocyte it, provided that the macrophage has not reached its maximum capacity (determined by a phagocytic index [72,73,74]). Moreover, we tested multiple values for the phagocytic fraction, i.e., the fraction of macrophages that is able to clear senescent cells.

#### 4.3.3. Migration

Contrary to the static nature of our previous work [21], this model features moving agents and mechanical interactions. In fact, moving to a finer scale allowed us to simulate cell migration while we relied on the implementation of the mechanical forces from a default BDM operation for the repulsion of cells in close proximity. We implemented two types of cell migration (none of them allows the cells to escape the alveolus they belong to), namely random movement and neighbourhood-informed migration.

We assumed that macrophages, mesenchymal cells, and damaged and senescent AEC2 would randomly move within the alveolus to either patrol the alveolar space (M1 and M2 macrophages), restore the ECM reserve (fibroblasts and myofibroblasts), or spread the damage while triggering the immune system (damaged and senescent AEC2). Within this group, we further distinguished cells that move at every time step from those that do not. Specifically, macrophages and mesenchymal cells in our model move constantly to perform the aforementioned activities. At every time step, a random direction (i.e., an angle in the interval (0, 2π)) is drawn from a uniform distribution and rotation matrices are used to compute the new position of the cell given its speed and the length of the time step. Moreover, macrophages that have exceeded their phagocytic capacity are not able to move anymore (see the section Damage spreading and clearance). On the contrary, damaged and senescent AEC2 might lose the ability to migrate properly due to radiation-induced injury. Therefore, at each time step, Δθ and Δϕ (where θ and ϕ represent the polar and azimuthal position of the cell) are drawn from a uniform distribution within a small range. We simulated casual absence of motion by including 0 in the range.

We implemented neighbourhood-informed migration to simulate AEC1 and AEC2 movements toward areas with low epithelial population density. In fact, AEC2 cells must migrate to regions depleted from AEC1 before starting the repopulation process, while AEC1 spread to maximize the alveolar gas-blood exchange capabilities [75,76]. We modelled these mechanisms as follows (no assumption was made on the intercellular communication and we used the inverse-square law to model general signalling):The epithelial cell detects all its neighbours within an alveolar radius distance in the alveolus it belongs to (AEC1 detect other only AEC1, while AEC2 detect both AEC1 and AEC2);For each of the detected neighbours, the distance from the acting cell is measured;Using the distances from the step above, the cell determines the centre of mass of its neighbours, where each distance is weighted with the inverse of its squared normalized value;The cell migrates in the opposite direction from the location of the centre of mass, given a constant migration speed and the length of the simulation step.

#### 4.3.4. Proliferation

Healthy AEC2 and fibroblasts are the only cell types that can (symmetrically) proliferate in our model. Due to their ability to act as stem cells, AEC2 proliferate to repopulate alveoli depleted from AEC1. In fact, it was shown that hyperplastic AEC2 are found in damaged alveoli [5]. We modelled AEC2 hyperplasia by providing the cells with the ability to check the state of their neighbours within an alveolar radius distance, which was made possible by the increase in the resolution with respect to our previous work [21]. If the number of AEC1 in the neighbourhood falls below a threshold (which we set as 20% less than the average number of AEC1 in homeostatic conditions in the monitored area), the AEC2 doubles its proliferation rate. Although more complex models of lung cells repopulation after radiation-induced damage have been developed [77], our models seek to replicate the ability of AEC2 cells to quickly and sensitively respond to local depletion of AEC1 by making use of a simple threshold mechanism. Fibroblasts, on the other hand, proliferate during the inflammatory process. We modelled their proliferation as in [21], with a rate that depends on TGFβ_active_, IL13, and FGF2, the latter being present only in the presence of senescent AEC2 [78,79,80].

#### 4.3.5. Differentiation

Differentiation is a process that involves changes in a cell’s phenotype and leads to the acquisition of new capabilities while others are lost. We mimicked this process by changing a cell’s type, removing its old behaviours, and assigning new ones. The differentiation is a behaviour itself, and in the following, we detail those that might take place in our simulations. Unless otherwise stated, the differentiation was modelled as in [21].

We assumed that monocytes entering the alveolar space would immediately differentiate into pro-inflammatory M1 macrophages [81]. On the contrary, M1 macrophages have a constant probability of turning into M2 macrophages at each time step (tuned such that their number in homeostasis is constant), while the probability that M2 macrophages differentiate into M1 depends on the local concentration of TNFα [82] (which affects it positively, though this process is rarer than the opposite one).

Fibroblasts differentiate into myofibroblasts that collaborate in the secretion of the ECM [19,20]. Although the myofibroblasts are almost absent from the alveolar space in homeostatic conditions, we assumed an initial level of inflammation (as was done in [46]) and tuned the differentiation rate so as to maintain a constant number of both fibroblast and myofibroblasts in homeostasis. This transition is then positively influenced by pro-fibrotic M2 macrophages through the secretion of PDGF and TGFβ_active_ [83].

Finally, AEC2, which act as a reservoir for the terminally differentiated AEC1, following migration and proliferation (as described in the previous sections), alter their phenotype at a constant rate [76,84].

#### 4.3.6. Apoptosis

We distinguish between constant and variable apoptosis. Cells that undergo constant apoptosis (AEC1, macrophages, and mesenchymal cells) are removed from the simulation at a constant rate. At each time step, each cell draws a random number from a uniform distribution, and if this number is smaller than the apoptosis rate, the removal is performed. On the contrary, variable apoptosis depends on external factors. As reported in the literature [75], healthy AEC2 increase their apoptotic rate if the number of surrounding AEC1 exceeds the homeostatic value. We modelled this behaviour as the adaptive proliferation (see Section 4.3.4), doubling the AEC2 apoptotic rate when the threshold (which we set as 20% more than the average number of AEC1 in homeostatic conditions in the monitored area) is exceeded. Finally, we assumed that damaged and senescent AEC2 could not undergo apoptosis due to the short duration of the state (the former) and the phenotypic resistance to it (the latter, as shown in the literature [85]), respectively.

### 4.4. Initial Conditions

#### 4.4.1. Homeostasis

As described in our previous work [21], we used a trial-and-error approach to tune the model parameters in homeostatic conditions. Except for the initial cell numbers, the size of the geometric frame, and the diffusion parameters of the extracellular substances (see Section 4.2), we calibrated the secretion, proliferation, apoptosis, and differentiation rates to (i) match homeostatic experimental values reported in a previous model of IPF [46] and (ii) keep the number of cells constant (i.e., within a reasonably small range of variation) for multiple weeks. Plots of the cell numbers and substances concentration are provided in the Appendix A.

#### 4.4.2. Radiation-Induced Damage

The radiation damage in our model was simulated as depletion of healthy AEC2. To replicate heterogeneous damage, depleted fractions for each alveolus were drawn from Poisson distributions centred on the average number of injured cells for a given dose. For each simulated dose, we estimated the surviving fraction by fitting experimental data from a study on rats AEC2 [47] (for doses < 15 Gy, due to its questionable validity at higher doses [86]) using the linear quadratic (LQ) model. To compensate for the lack of human pneumocyte data in our work, we limited our references to in vivo studies on AEC2 and neglected in vitro experiments on A549 and H460 cell lines. According to the LQ model (the most commonly used in radiation biology [87]), the survival probability (*S*) for a cell exposed to a single fraction of dose *D* is given by:(6)$$S=e^{-\alpha D-\beta D^{2}}$$
where the *α* and *β* parameters characterize the radiosensitivity of a cell line. Figure 9 shows the fit curve and parameters used in our simulations.

According to our fit, the value of *β* is one order of magnitude smaller than *α* (or, equivalently, the *α/β* ratio is large), in agreement with results from the radiosensitivity of multiple mouse strains [26].

## 5. Conclusions

To our knowledge, our work represents the first attempt to mechanistically model RILF using the agent-based modelling approach. We showed that the time evolution of the ECM concentration in our alveolar duct model could replicate early and late-response patterns observed in patients treated with RT. Moreover, the ECM concentration peak in the early response of our model is reached after 3-4 months from the irradiation, as found by previous studies. Further, we measured the FSU survival rate as a function of the dose and observed discrepancies at higher doses when fitting the data points with the critical volume model. Our RILF severity index, a surrogate measure of the disease severity based on the work done by Zhou et al., is in agreement with the experimental results shown in their mice model of RILF [37]. Finally, we highlighted the role of the senescent AEC2 as the main triggers of the RILF by reporting lower survival curves for the alveoli when increasing the initial fraction of senescent cells.

Future developments will focus on replicating a bigger portion of the lungs and coupling a radiation transport model with the aim of making voxel–voxel comparisons.

## Figures and Tables

**Figure 1 ijms-23-13920-f001:**
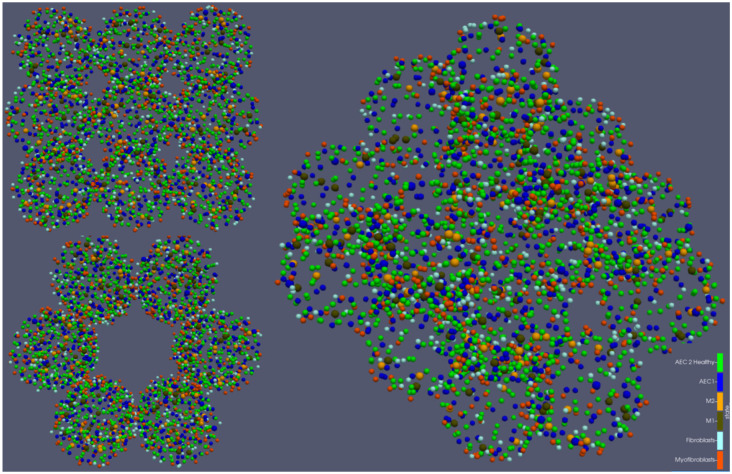
Visual output of the 3D ABM of a human alveolar duct. The structure is made up of 3 stacked layers, and each layer consists of 6 tangent alveoli. The centres of the alveoli are located on equidistant circles with a radius equal to the duct radius. Cells are represented as coloured spheres, with green = AEC2 cells, blue = AEC1 cells, brown = M1 macrophages, orange = M2 macrophages, light blue = fibroblasts, red = myofibroblasts.

**Figure 2 ijms-23-13920-f002:**
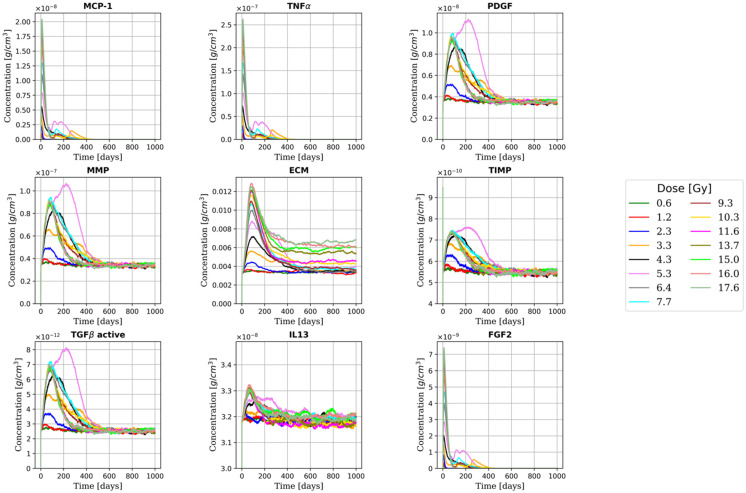
Time evolution of the extracellular substances for multiple doses. Peaks in the MCP-1, TNFα, and FGF2 concentration, secreted by senescent AEC2, are observed immediately after the irradiation (see [38]). As shown in [39], a dose-dependent increase in the concentration of TGFβ (and the other macrophage-derived cytokines) is observed, followed by a restoration of the homeostatic levels in the late fibrotic phase as the senescent cells are fully cleared. A characteristic two-component pattern, where an initial dose-dependent peak is followed by a settlement, can be seen in the ECM concentration (as observed in [34]). The ECM, secreted by both fibroblasts and myofibroblasts, returns to homeostatic levels in the late phase only for doses ≤5 Gy.

**Figure 3 ijms-23-13920-f003:**
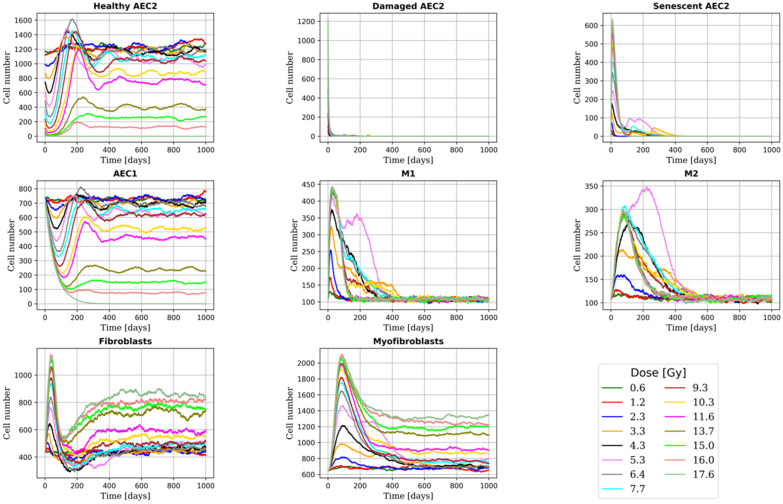
Time evolution of the total number of cells for multiple doses. As shown in [5], the AEC2 population is fully restored at low doses, while above 5 Gy, the increase in the number of senescent AEC2 (which grows with the dose) leads to the complete depletion of the healthy ones in a fraction of the alveoli (see Section 2.2). Consequently, the total number of healthy AEC2 decreases and then settles, followed by a decay in the AEC1 population. M1 macrophages, triggered by the damaged epithelium, accumulate proportionally to the number of senescent AEC2 and later on differentiate into M2 macrophages. In the later stage of fibrosis, following the clearance of the senescent cells, a decrease in the macrophage population is observed [40,41]. Finally, the population of mesenchymal cells, whose proliferation is stimulated by macrophages and senescent AEC2 in the early phases, settles at higher levels (with respect to homeostasis) in the alveoli depleted of healthy AEC2.

**Figure 4 ijms-23-13920-f004:**
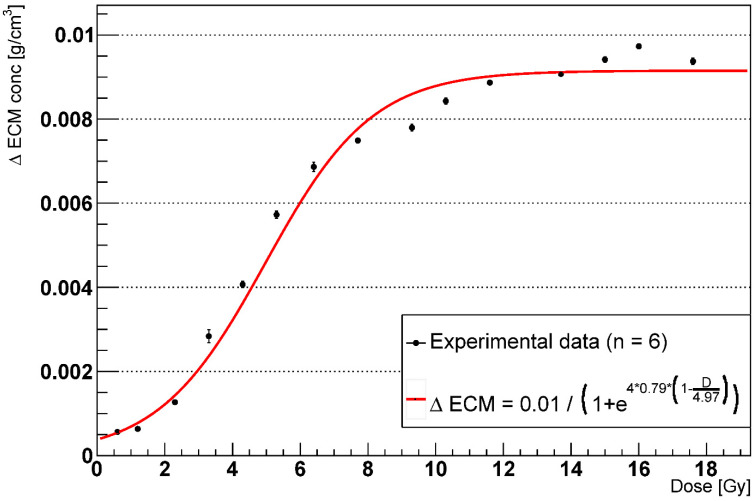
Dose-response curve for the early ECM concentration increase after 3 months with phagocytic fraction = 100% (n = number of experiments).

**Figure 5 ijms-23-13920-f005:**
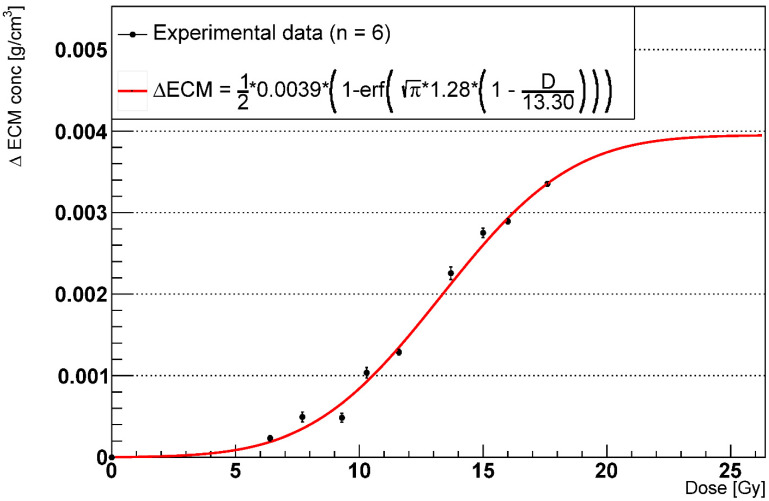
Dose-response curve for the ECM concentration increase after 1000 days with phagocytic fraction = 100% (n = number of experiments).

**Figure 6 ijms-23-13920-f006:**
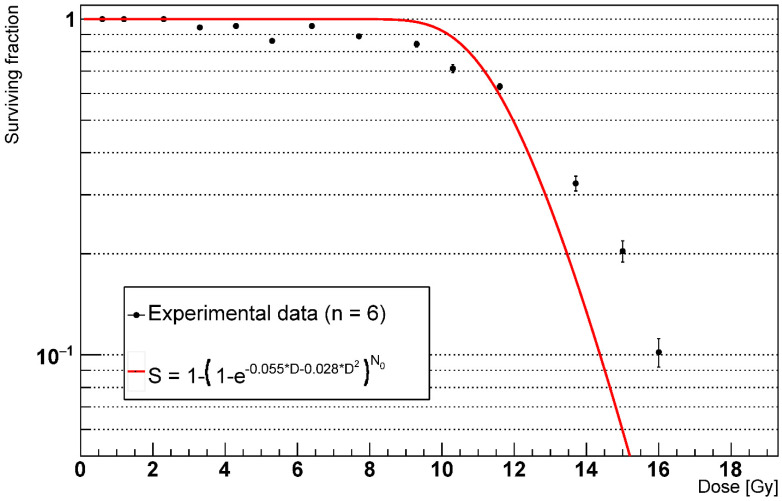
Alveoli survival after 1000 days with phagocytic fraction = 100% (n = number of experiments).

**Figure 7 ijms-23-13920-f007:**
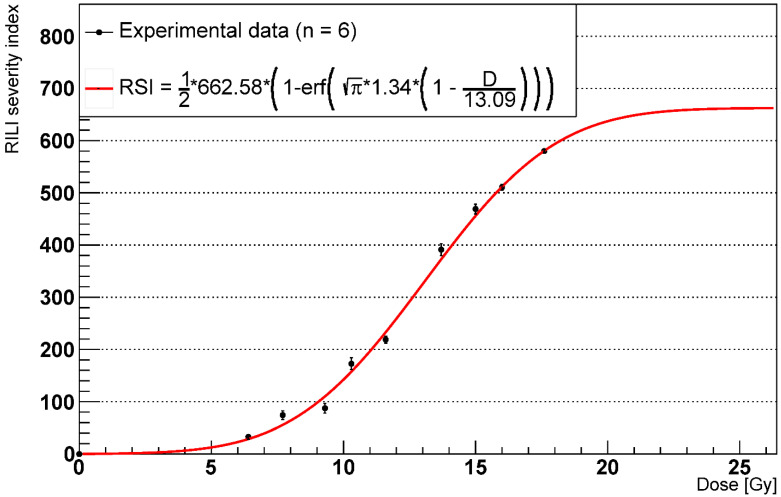
RILI severity index after 1000 days with phagocytic fraction = 100% (n = number of experiments).

**Figure 8 ijms-23-13920-f008:**
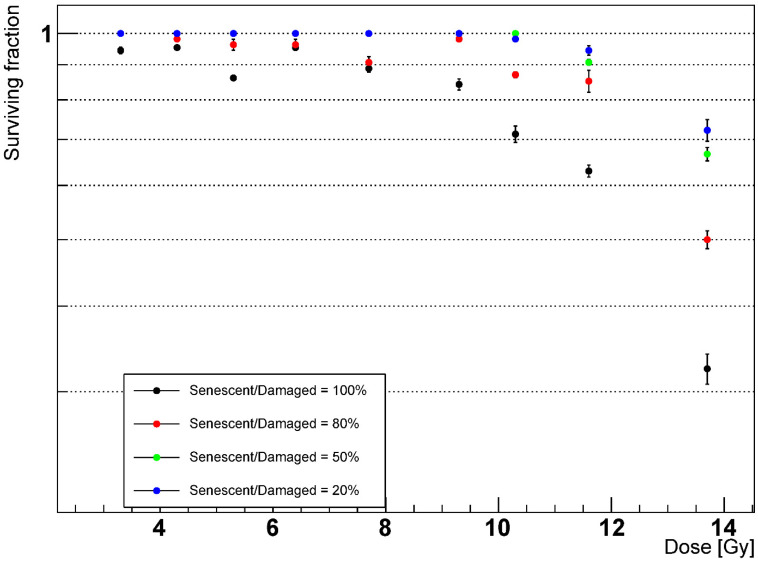
Alveoli survival after 1000 days with phagocytic fraction = 100% and different senescent to damaged AEC2 ratios. Senescent/Damaged = x% means that, after the irradiation, a fraction x of the healthy AEC2 shifted to the senescent state, while the remaining 1 − x fraction underwent apoptosis.

**Figure 9 ijms-23-13920-f009:**
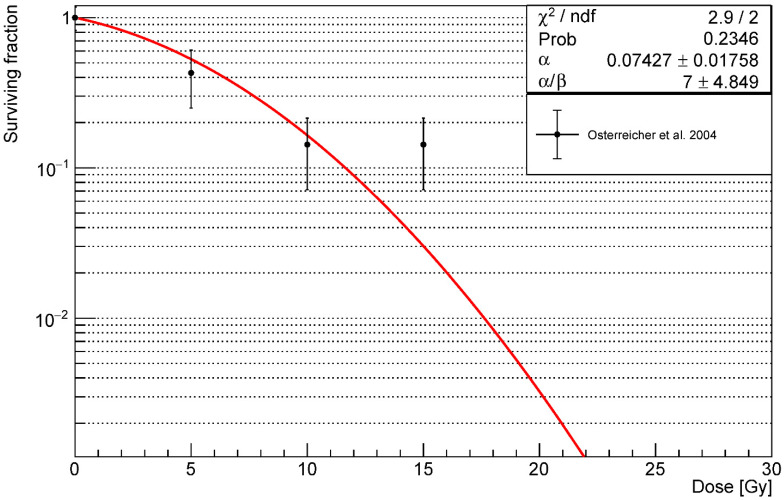
Experimental data of rats AEC2 survival from [47]. Data were fitted using the LQ model (in red) and the curve was used in the ABM to convert the simulated damaged fractions into the corresponding delivered doses.

**Table 1 ijms-23-13920-t001:** Cell types and numbers.

Category	Type	Initial Number per Alveolus	Source
Epithelial	AEC1	41	[60]
	AEC2 healthy	69	[60]
	AEC2 damaged	0	
	AEC2 senescent	0	
Macrophage	Macrophage M1	6 ^1^	[60]
	Macrophage M2	6 ^1^	[60]
Mesenchymal	Fibroblast	24 ^2^	[46,60]
	Myofibroblast	36 ^2^	[46,60]

^1^ Contrary to our previous model, we include only alveolar macrophages (M1 and M2) and neglect interstitial macrophages. We assume that M1 and M2 are equally distributed in homeostatic conditions. ^2^ See our previous work [21].

## Data Availability

All the relevant data is contained within the article and its Appendix A.

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
