# Peer review of "An Agent-Based Model of Radiation-Induced Lung Fibrosis"

_ijms, 2022, doi:10.3390/ijms232213920_

Round 1

Reviewer 1 Report

The manuscript is interesting and written in good English. However, it is mainly an extension of the previous work, by the same authors (ref 22) and with a very similar title.

Furthermore, I believe that the introduction is too long, while there are no conclusions which are not entirely clear from the discussion of the authors.

Author Response

  • Referee 1
    1. We do agree and indeed the titles of the current and previous manuscript look very similar. However, with our previous work we replicated the onset of idiopathic pulmonary fibrosis and we neglected radiation (term that was added in the new manuscript’s title) damage-specific tissue responses. Moreover, in our last work we downscaled the model to focus on inter- and intra-alveolar interactions and highlighted the role of senescent epithelial cells. This will allow, in the future, to incorporate additional factors during the treatment that have been largely ignored in the past in computational models. Our new model therefore constitutes a platform to formulate, test and compare different hypotheses on the impact of treatment options, such as for instance the combination of RT with specific drugs, which has the potential to support and guide clinical decision-making (see, for example, Demetriades et al., Interrogating and Quantifying In Vitro Cancer Drug Pharmacodynamics via Agent-Based and Bayesian Monte Carlo Modelling, Pharmaceutics, 2022). Finally, we implemented senescent cells-clearance mechanisms in the macrophages and matched the resolution of the diffusion grid (which is used to track the concentration of the substances) with the average size of a computational tomography voxel (~ 0.5 mm).
    2. Thank you for your comment. Non-relevant details were removed from the Introduction and the overview on the structure of the lungs’ airways was shortened and moved to the Methods where the geometric framework of the model is described (rows 639 to 641). We have also included a figure to improve the clarity and structure of the previously text-only Introduction (below row 124). Further, after the Methods section, we added the Conclusions where the key points from the Discussion were collected and the few lines regarding future developments were moved (row 903).

Reviewer 2 Report

              This study developed a cell-based model to predict lung fibrosis, but no NTCP model has been developed to predict clinical results. The authors claim that current NTCP models rely on clinical experience. Meanwhile, an agent-based model of radiation-induced lung fibrosis was simulation only. I think the decisive difference between real-world and simulation is whether or not the presence of a tumor (i.e., tumor microenvironment) is taken into account. Such tumor microenvironment causes the ECM re-modeling, immune response, and non-targeted effects on the normal tissues. This phenomenon should be considered into the model. So how do you link events (ⅰ)-(ⅲ) to the presence of a tumor?

Specific comment

-          What does ECM in ECM concentration refer to? It is clude and confusing.

-          The fraction of phagocytic macrophages is also confusing. Is this model parameter? Or measurement values? If the latter, how did you measure it?

-          The definition of cell survival and radiation damage should be clearly defined i.e., clonogenic potential and γH2AX foci etc.

-          Equation (3) looks like a target theory. Why did you use the LQ model?

-          Figures 3-8 need to be improved about the vertical axis and orientation.

Author Response

  • Referee 2
    1. Your observations regarding the lack of a new NTCP model and the tumor microenvironment are absolutely reasonable. In fact, the aim of the whole project is to ultimately develop a new NTCP model for RILF, but it’s a multi-step process and the goal of our manuscript is to report the current state of the ABM which builds on our previous work. We added new radiation damage-specific response mechanisms and lowered the scale to be able to implement cell behaviors and focus on post-irradiation indirect damages besides the direct ones. As of now we simulate the onset of RILF in a healthy substructure of the lungs (an alveolar segment), which we suppose is located far enough from the tumor microenvironment so that it’s not influenced by it, yet still close enough to be irradiated and damaged by the radiation. Therefore, we simulate the onset of RILF caused uniquely by the damage to healthy alveoli, as reported in the literature for mice (see, for example, Citrin et al., Role of type II pneumocyte senescence in radiation-induced lung fibrosis. J Natl Cancer Inst, 2013, or Zhou et al., Quantitative assessment of radiation dose and fractionation effects on normal tissue by utilizing a novel lung fibrosis index model. Radiat Oncol, 2017).
    2. Thank you for your comment on the ECM concentration, we added its definition in the Results before its first appearance (rows 384 to 386). ECM concentration refers to the average concentration of the extracellular matrix in g/cm3 across the whole simulation space. Since we couldn’t measure the density change in Hounsfield Units (as was done by previous studies on patients affected by RILF), we decided to use a surrogate the concentration of the ECM given the existing relationship between the attenuation coefficient and the density.
    3. Regarding the fraction of phagocytic macrophages, we used it as a model parameter. In particular, we set it at the beginning of each simulation so that only a fraction of the macrophages was able to clear senescent cells. For each newly recruited macrophage, there was a probability of it being phagocytic or not. We tested different combinations of phagocytic fractions (60%, 80% and 100%) and phagocytic index (defined as the number of cells that one macrophage can clear during its life cycle) and showed the output for the models that led to the best results. In the revised manuscript we specified that the fraction of phagocytic macrophages is a model parameter at the beginning of section 2.1. (row 322).
    4. With our model at its current (early) stage we assume that each cell that is damaged by irradiation can either undergo apoptosis or become senescent. In either case, the cells can no longer proliferate or differentiate. Therefore, the survival curves that we provide in our manuscript show the fraction of cells that have clonogenic potential. Given the simplicity of our model, we assume (as of now) that all intracellular damages have the same entity and therefore a cell can be either damaged by irradiation or not. We don’t take into account repair mechanisms and don’t estimate the number of DSB per cell. Further, for the sake of simplicity, we assume that only AEC2 can be damaged by irradiation. In the revised manuscript we have added the details reported above in section 2.2. (rows 426 to 430).
    5. Thank you for your comment regarding equation (3). Given its wide adoption for parallel organs (such as the lungs), we decided to fit some of the results of our model with the critical volume model for NTCP (see Stavrev et al., Critical volume model analysis of lung complication data from different strains of mice. Int J Radiat Biol, 2005 or Niemierko et al., Modeling of normal tissue response to radiation: The critical volume model. Int J Radiat Oncol Biol Phys, 1993). In particular, we assumed the alveoli as functional subunits of the lungs. Equation (3) does indeed look like the target theory, but its meaning is actually different. With it, we want to express the probability of survival for a whole functional subunit, i.e. an alveolus, but no subcellular compartment is taken into account. Given the definition of FSU survival in the critical volume model, we assume that an FSU survives after the irradiation if and only if no stem cell is left alive. In fact, even a single stem cell (an AEC2 for the alveoli) would be able to restore the FSU. Therefore, we express that probability as 1 minus the probability that the whole FSU is killed (either undergoing apoptosis or becoming senescent). As mentioned before, the latter happens only if all the cells that belong to the FSU are inactivated and thus we want the joint probability that all the NAEC2 are killed (= (Pkill, 1 cell)^(NAEC2)). Finally, for a single cell, the critical volume model assumes that the LQ model can be applied and applies it to estimate the survival probability of a single cell. In the revised document we clarified what aforementioned in section 2.2. (row 439).
    6. We sincerely apologize for the flipped and shrunk figures. Those were unfortunately resized and moved during the conversion to PDF format and subsequent upload, but we’ll make sure that the same won’t happen again in the future.

Round 2

Reviewer 1 Report

I think that the manuscript can be accept in present form

Author Response

We thank the reviewer for his positive assessment of our revision.

Reviewer 2 Report

I think you revised the point clearly, but Figures are not improved. I strongly recommend remaking them.

Author Response

We thank the reviewer for his positive review. As to the figures, there seem to be a problem in the PDF conversion, we are very sorry for that. We uploaded separately the original figures that are actually in HighRes and suitable for publication. Apologies for the inconvenience.

Round 3

Reviewer 2 Report

I confirmed that the figures were improved.